# Hyaluronic Acid/Bone Substitute Complex Implanted on Chick Embryo Chorioallantoic Membrane Induces Osteoblastic Differentiation and Angiogenesis, but not Inflammation

**DOI:** 10.3390/ijms19124119

**Published:** 2018-12-19

**Authors:** Laura Cirligeriu, Anca Maria Cimpean, Horia Calniceanu, Mircea Vladau, Simona Sarb, Marius Raica, Luminita Nica

**Affiliations:** 1Faculty of Dental Medicine, Department 3/Odontotherapy and Endodontics, Victor Babes University of Medicine and Pharmacy Timisoara, Timisoara 300041, Romania; lauracirligeriu@yahoo.com (L.C.); luminita.nica98@yahoo.ro (L.N.); 2Department of Microscopic Morphology/Histology, Angiogenesis Research Center, Victor Babes University of Medicine and Pharmacy Timisoara, Timisoara 300041, Romania; simona_sarb@yahoo.com (S.S.); raica@umft.ro (M.R.); 3Faculty of Dental Medicine, Department 1/Parodontology, Victor Babes University of Medicine and Pharmacy Timisoara, Timisoara 300041, Romania; horia_calniceanu@yahoo.com; 4Vladau Dental Clinic, Brasov 500173, Romania; office@vladau.ro

**Keywords:** bone substitutes, chorioallantoic membrane, RUNX 2, SPARC, BMP4

## Abstract

Microscopic and molecular events related to alveolar ridge augmentation are less known because of the lack of experimental models and limited molecular markers used to evaluate this process. We propose here the chick embryo chorioallantoic membrane (CAM) as an in vivo model to study the interaction between CAM and bone substitutes (B) combined with hyaluronic acid (BH), saline solution (BHS and BS, respectively), or both, aiming to point out the microscopic and molecular events assessed by Runt-related transcription factor 2 (RUNX 2), osteonectin (SPARC), and Bone Morphogenic Protein 4 (BMP4). The BH complex induced osteoprogenitor and osteoblastic differentiation of CAM mesenchymal cells, certified by the RUNX2 +, BMP4 +, and SPARC + phenotypes capable of bone matrix synthesis and mineralization. A strong angiogenic response without inflammation was detected on microscopic specimens of the BH combination compared with an inflammatory induced angiogenesis for the BS and BHS combinations. A multilayered organization of the BH complex grafted on CAM was detected with a differential expression of RUNX2, BMP4, and SPARC. The BH complex induced CAM mesenchymal cells differentiation through osteoblastic lineage with a sustained angiogenic response not related with inflammation. Thus, bone granules resuspended in hyaluronic acid seem to be the best combination for a proper non-inflammatory response in alveolar ridge augmentation. The CAM model allows us to assess the early events of the bone substitutes–mesenchymal cells interaction related to osteoblastic differentiation, an important step in alveolar ridge augmentation.

## 1. Introduction

Alveolar ridge augmentation represents a preliminary step in dental implant procedures. Several studies and highly heterogeneous data have been reported regarding the techniques applied and biomaterials used for alveolar ridge augmentation. The most controversial issue in the field is the nature of the biomaterials used for augmentation [1,2,3], and also their effects on host tissues. Some augmentation techniques are complex and involve invasive procedures not well tolerated by patients [4,5]. As a perspective for alveolar ridge augmentation, 3D printing allows for the production of biodegradable and bioresorbable bone scaffolds with patient-specific dimensions using computer-aided design [6,7]. 

Despite the fast development of alveolar ridge augmentation techniques, bone substitutes are still largely used for this procedure. Bone substitutes commercially available as Bio-Gen or Bio-Oss are spongy or cortical bone of equine or bovine origin, fully enzyme deantigenised, able to be resuspended in sterile saline solution and to be used as a scaffold for alveolar bone augmentation. The osteoconductive properties of these bone substitutes are widely recognized, but the most frequent criticism of these materials is that they only have osteoconductive properties, but not osteoinductive properties [8]. In contrast with these data, a few recent reports highlighted the osteoinductive properties of a calcium-phosphate complex (also found in bone substitutes) based on several pathways, as stimulation of mesenchymal stem cells recruitment and commitment through an osteoblastic lineage, together with a strong angiogenic process [9,10]. 

Several animal experimental models are used to test different bone substitutes and their interactions with alveolar bone and periodontal tissues. Tests usually performed on dogs, rabbits, and rats have the main disadvantage of lacking early assessment of the effects of bone substitutes on periodontal tissues. Additionally, in this experimental model, the interaction between mesenchymal stem cells and bone substitutes is hard to quantify. This happens because the biopsies are harvested three to six months after experimental alveolar ridge augmentation [11,12], and early events of bone substitute integration are missed.

Chick embryo chorioallantoic membrane (CAM) is well known as a powerful experimental tool for the study of normal and pathologic angiogenesis, because of its high vascular network [13,14]. Additionally, CAM is an embryonic tissue lacking an immune system and having mesenchymal cells with stem like potential able to differentiate depending on a specific microenvironment [15,16]. Based on the previously described features, CAM is a reliable in vivo model for testing the behavior of normal and pathologic tissues (as cultured cells or malignant tumors) [17,18], of different drugs and antibodies [19,20], or of a variety of biomaterials implanted on its surface [21,22]. There is minimal data available regarding bone implants on CAM [23,24]. Despite this, there is some testing of the vessel acquisition ability of different implanted biomaterials, such as 3D printed hydroxyapatite scaffolds [25] or nanoparticles [26]. CAM is even less used as an experimental model in dental medicine research. Dental stem cells angiogenesis [27], the behavior of enamel implanted on CAM [28], or CAM testing of a limited number of biomaterials used in dentistry [29,30], are few examples of CAM use in dental research. No data on bone substitutes tested on CAM are available in the literature. 

Based on the versatility of CAM given by its immature mesenchymal structure and its ability to have no immune system, we propose here CAM as a model for testing one of the bone substitutes usually used for alveolar ridge augmentation. We used three different resuspension media to assess if different resuspension media may influence bone substitute behavior. The main objectives of the present study were focused on: (1) The identification of a possible osteoinductive effect of the bone substitute on CAM mesenchymal cells, (2) the assessment of inflammatory response induced by the CAM–bone substitute interaction, and (3) the effects of the bone substitute on CAM vessels and angiogenesis. 

## 2. Results

### 2.1. Microscopic Assessment

The specimen survival rate was over 95%, and those who died before the end of the experiment were most likely influenced by daily manipulation rather than by the implant. 

Significant macroscopic changes were dynamically observed during the experiment, as shown in Figure 1.

Hyaluronic acid (H) alone induced a discrete increase of CAM vascularization, which started early on day 1 (Figure 2a); was maximal on day 2, when we observed discrete suffusion bleeds (Figure 2b); and then on day 4 to 7, the samples did not show significant changes compared to day 1 and 2 (Figure 1c,d). Sterile saline solution (S) was not accompanied by vascular changes or other particular aspects of CAM, this being macroscopically similar to normal CAM for each developmental stage (Figure 1e–h). On day 7, the BH complex induced a persistent compact layout of interlaced bone lamellae with obvious spaces between them, similar to those of the alveolar bone. Additionally, the BH complex stimulated blood vessel development around the implant, highlighted by an intense angiogenic response. 

BSH produced different specific changes compared to the previous compounds. The bone particles remained dispersed within the silicone ring, where they were implanted (Figure 1m,n). The vascular reaction was present amongst the bone particles, but it was particularly evident around the silicone ring, the vessels being specifically arranged in the form of “wheel spokes” (Figure 1n,o).

CAM biopsies harvested on day 13 were histopathologically and immunohistochemically evaluated. The microscopic examination of the H samples highlighted a large number of blood vessels, confirming a vascular reaction in the CAM, while for the samples treated with saline solution (S), the number and distribution of the blood vessels was similar to that of normal CAM.

The morphology of the mesenchymal cells from CAM chorion in H-treated specimens was significantly changed compared with control specimens. We observed that CAM chorion cells became elongated with a spindle like shape and a tendency to be grouped as compact structures just below CAM epithelium. The same aspect of stromal cells reorganization was also observed in BH treated specimens.

For the BSH samples, a large inflammatory infiltrate was detected around the implant. At the end of the experiment, resorption areas prevailed, detrimental to the compact areas. In compact areas, resuspended bone cells showed an intense acidophilic cytoplasm and were poorly interconnected. 

At the end of the experiment, the BH complex showed the lowest resorption rate. Around BH samples, there was an angiogenic response in the absence of inflammation.

Microscopically, on specimens treated with BH complex, compact structures were observed, composed of tightly interconnected cells able to create a continuous, well-defined layer, tightly attached to the internal surface of the CAM (Figure 2a). New blood vessels densely packed around the BH implant were observed, and they already penetrated the edges of the implant and were perfused. No inflammation has been detected on BH specimens (Figure 2b). Stellate shaped mesenchymal cells of the normal CAM chorion were replaced by highly mitotic cells with spindle morphology (Figure 2c,d). This spindle appearance of chorionic cells suggested to us a differentiation through an osteoblastic lineage. Based on this evidence, immunohistochemistry was performed. 

### 2.2. Immunohistochemistry Evaluation

A panel of antibodies involved in osteoblastic differentiation (RUNX2), bone matrix synthesis (SPARC), and ossification (BMP4), were applied and different results obtained.

Mesenchymal cells from samples treated with hyaluronic acid (H) alone were negative for RUNX2, BMP4, and SPARC (Figure 3a, b). Specimens of CAMs treated with saline solution (S) did not show significant changes compared to H-samples, except for inflammatory cells where SPARC and BMP4 immuno-expression was scattered and inconsistently positive.

BS samples showed a positive but inconsistent reaction for RUNX2, and SPARC was not expressed in these types of samples, nor BMP4. The BSH complex was also reflected in the expression pattern of the three markers included in the study, and completed the morphological characterization of BSH-treated samples. Thus, RUNX2 was expressed in the mesenchymal cells of the CAM adjacent to the implant. The expression was mild regarding the density and intensity of immunohistochemical reaction in the implant areas (Figure 3c). The stromal cells in which the implant induced RUNX2 expression were dispersed throughout the CAM chorion area, keeping the organization of a normal CAM (Figure 3d). Regarding SPARC, the reaction was heterogeneous among specimens. If for some BSH specimens this reaction could not be observed, other BSH specimens showed an intense SPARC reaction in both stromal cells and endothelial cells, most likely induced by associated inflammation (Figure 3e). BMP4 was inconstant and focal positive within the BSH implant, with a distribution suggesting its positivity only at the level of invasive vessels (Figure 3f). 

The BH complex induced significant changes, both morphological (previously described) and molecular, characterized by the expression of RUNX2, BMP4, and SPARC. A particular aspect was encountered in the case of the BH complex and was represented by stratification of RUNX2 positive cells (Figure 4a).

The stratified organization induced by the BH implant was characterized by heterogeneity in the distribution of RUNX2 positive cells. The highest density was recorded inside the BH implant, giving the appearance of a compact structure (Figure 4b). 

The condensed stroma adjacent to the BH implant represented the second layer composed of spindle cells with positive nuclei for RUNX2 (Figure 4c). Cells of the second layer were dispersed compared to those of the first layer, but the density of RUNX2 positive cells was still high. The third layer was composed of mesenchymal like cells strongly positive for RUNX2 (Figure 4d). The limits between these three layers were distinct (Figure 4e,f).

The other two markers were strongly positive for the BH treated samples too. SPARC was expressed in all three layers with cytoplasmic expression and some scattered nuclear expression, more intense in the implant areas and moderately weak in the CAM modified chorion (Figure 5a–c). BMP4 had the same distribution with the particularity that its expression was heterogeneous in CAM layers around the implant (Figure 5d–f). 

A comparative assessment of the results is summarized in Table 1. 

## 3. Discussion

Alveolar ridge augmentation was developed at the beginning of the 1980s by Tatum and Boyne, and made possible the rehabilitation of implantology and increased the success rate of the dental implants by restoring alveolar bone atrophy and preparing it for the future dental implant [31,32]. The alveolar bone augmentation technique is currently an efficient procedure with a very solid scientific and practical basis, but is also dependent on the regenerative capacity of the alveolar bone in patients who initially did not have anatomical or biological contraindications for this type of surgery [33,34].

The diversity of biomaterials used, respectively organic or inorganic materials of animal or synthetic origin, as well as the heterogeneity of the resuspension media of these materials, are the basis of the controversial results in the literature, as well as the failure of dental implants following these interventions. Moreover, the characterization of the early stages of bone augmentation behavior, both at the morphologic and molecular level, are rare and incompletely studied. It is well known that any natural bone material used as xenograft in alveolar ridge augmentation has osteoconductive properties, and few of them have osteoinductive effects. 

The alveolar bone has a particular histological structure, consisting of compact bone and spongy bone, which is why the materials used should contain both cortical bone and spongy bone particles.

Experimental studies for alveolar bone substitutes are extremely limited at this time. Relatively recent studies have tested the efficacy of the BioGen Bioteck bone equine implant for experimental fractures induced in rabbits, sheep, or rats [35,36,37,38], but alveolar bone augmentation with such biomaterials has been poorly studied, especially on canine and rabbit models [39,40,41,42,43]. The evaluation and validation of the data obtained in these studies has been achieved mostly by radiology, and only in few cases by histological and morphometric methods. Much less studied were the molecular factors involved in osteoconductivity, osteoinduction, and the osteoprogenitor features of the implants. 

The experimental model based on the chorioallantoic membrane allows the observations of bone substitute effects starting from the early stages, an aspect that cannot be studied in other experimental models, due to the failure of daily microscopic assessment. Other experimental models do not allow the dynamic follow-up of the bone implant and adjacent stroma interactions, nor the molecular differences involved in these stages of the implant. Bone grafts on CAM are rarely cited in the literature. Most bone graft models use bone fragments stored in tissue banks [44], and so far no study that has tested the equine bone from the Bio-Gen preparation on the CAM model has been cited in the literature. This, together with the patient’s variable clinical response to the Bio-Gen implant, was the main reason for initiating this study. The main objective pursued in these experiments was the bone-induced vascular reaction on CAM [45], or the cellular effects on stromal or implant components [22,45,46]. Among the first clinical studies using bone augmentation with the Bio-Gen implant are those of Piatelli (2002) [47] and Di Stefano (2009) [48], which also indicated that the vascular reaction is mandatory for bone regeneration, as well the expression of VEGF in the newly formed bone compared to the control group. Vascular reaction was also noted in our model, dependent on the resuspension medium of Bio-Gen bone particles. Our results confirmed the importance of the vascular network in the dynamic behavior of the Bio-Gen implant, but at the same time, it was pointed out that the development of the vascular network is dependent on the resuspension media. Based on the results previously described in this paper, the resuspension of bone granules in hyaluronic acid represents the optimal technique for the success of the implant, due to both anti-inflammatory and angiogenic effects that are well-known and cited in the literature [20]. A distinct aspect of the present experimental research is represented by the observation of the osteoinduction on CAM mesenchymal cells with a particular stratified distribution, as described in the results section. Our results for the CAM experimental model are in line with those described by Rachmiel and collaborators [49], who have recently characterized the alveolar bone formation steps after application of “bone distraction” techniques. The authors reported that the area of bone regeneration has three distinct zones, with vascular networks in-between necessarily interposed. Similar microscopic images were obtained in the present study on the group treated with the BH complex, which suggests the specific osteoinductive role of the BH complex in CAM mesenchymal cells. An original element of the present study is the RUNX2 evaluation of the implant site, showing that the density of RUNX2 positive cells is different in the three areas, an aspect not discussed in the above study. Rachmiel noted the importance of BMP2, and to a lesser extent BMP4, in the selection, induction, osteoinduction, and proliferation of osteoblasts at the site of bone regeneration. Our study has confirmed the importance of BMP4 in osteoblastic differentiation and the formation of the tri-laminated, layered bone regeneration using one type of bone substitute combined with hyaluronic acid (BH). The present study included SPARC evaluation, a protein that was not discussed by Rachmiel’s team in their study, and which is mandatory for the evaluation of collagen synthesis and control, but also for optimal bone mineralization,. We considered the assessment of SPARC in the characterization of the experimental model as an essential step, due to its dual role in bone mineralization and induction of the endothelial cell proliferation mandatory for osteogenesis. RUNX2, SPARC, and BMP4 are a panel of markers able to characterize the early stages of osteoinductive, osteoprogenitor, and osteoblastic effects of the bone substitutes combined with hyaluronic acid instead of saline solution or other combinations.

## 4. Materials and Methods

### 4.1. Bone Substitutes and Resuspension Media

For the purpose of this study we used granules of spongy and cortical bone of equine origin, fully enzyme deantigenised (B) (BIO-GEN^®^, Bioteck, Vincenza, Italy), and resuspended in three different excipients: Hyaluronic acid (H), (Hyadent BG, Bio Science, Dummer, Germany), sterile saline solution (S) (mostly used in clinical practice for bone granules resuspension), and a combination of hyaluronic acid and sterile saline solution (HS). Based on our previous observations regarding the diversity of the clinical response to the different types of excipients used for resuspending bone particles, we tested the chorioallantoic membrane reaction to bone granules (B) combined with saline (BS), hyaluronic acid (BH), and both hyaluronic acid and saline (BSH). Four types of resuspension media were tested on CAM: H, S, BH, and BHS. Details regarding these substances and their combinations are summarized in Table 2.

### 4.2. Chick Embryo Chorioallantoic Membrane Experimental Model

To establish the experimental model of CAM, 22 fertilized hen eggs were selected using the in ovo transillumination method. Eggs were sterilized and incubated for 72 h at 37 °C in a 60% humidity atmosphere. On the fourth day of incubation, a pointed puncture hole was formed at the narrow pole of the egg and approximately three milliliters of egg albumen was removed, followed by paraffin sealing of the hole. One day later, a shell window was performed for each specimen and viable embryos were selected. Chorioallantoic membrane was evaluated regarding its integrity and viability, and a silicon ring was applied on it. On the next day, the specimens were organized into five groups: One control group (with 2 eggs) and four test groups (with 5 eggs each). H, S, BH, and BHS were applied inside the silicon ring. The experiments ended on day 13 of incubation. For all procedures the ethical European rules regarding the use of animals for experimental purposes were respected. Because the CAM experimental method does not apply direct traumatic methods to chick embryos, and no pain occurs due to the lack of nerve fibers in the CAM, ethical approval was not mandatory for this type of experimental model. 

### 4.3. CAM Harvesting and Primary Processing for Microscopy and Immunohistochemistry

The experiments were stopped by in ovo fixation of CAM with 10% buffered formalin for 30 min. Then, CAM and the corresponding implants were removed and continued to be fixed for 24 h, followed by the paraffin embedding procedure. Three micrometer serial sections were performed, and routine hematoxylin and eosin stain was performed for histopathologic evaluation. Corresponding sections from each specimen were selected for immunohistochemistry. 

Immunohistochemistry helped us assess the osteoblastic commitment of mesenchymal cells, osteoblastic differentiation, and bone formation. Three different antibodies were selected. We used rabbit polyclonal anti RUNX2 (Runt-related transcription factor 2) antibody (Santa Cruz Biotechnology, dilution 1:100), well known as the key transcription factor associated with osteoblast differentiation; rabbit polyclonal anti SPARC (osteonectin) antibody (Abcam, dilution 1:100), a bone glycoprotein secreted by osteoblasts; and rabbit polyclonal anti BMP4 (bone morphogenic protein) antibody (Novus Biological, dilution 1:500), a polypeptide specifically involved in bone repair. Incubation with primary antibody for 30 min at room temperature was followed by the use of the visualization system Novolink Max Polymer/DAB. All immunohistochemistry steps were fully controlled using Max Bond Autostainer (Leica Microsystems, Medist Life Sciences, Bucharest Romania). 

### 4.4. Image Analysis and Data Acquisition

All CAM specimens were daily evaluated in ovo using an Axio CAM Stereo Microscope (Zeiss, Oberkochen, Germany), and pictures from different moments of the experiment were obtained and processed with ZEN software (version 2, Zeiss). Microscopic evaluation of the specimens was performed using an Axio Zoom 2 Observer Microscope (Zeiss).

## 5. Conclusions

This study is the first CAM experimental model testing biomaterials used in bone augmentation. Undifferentiated CAM mesenchymal cells can be considered as an in vivo platform of cells with a high potential of differentiation, similar to those found in alveolar bone, which are able to differentiate into the osteoblastic lineage. The BH complex induced CAM mesenchymal cells differentiation towards osteoprogenitor and osteoblastic lineage. Based on our experimental results, the BH complex is an ideal combination for alveolar ridge augmentation, because it does not induce inflammation, and its osteoinductive effect on mesenchymal cells was certified in the present study by RUNX2 +, BMP4 +, and SPARC + phenotype capable of bone matrix synthesis and mineralization. Additionally, our study supports the use of mesenchymal cells combined with bone substitutes and hyaluronic acid in alveolar ridge augmentation.

## Figures and Tables

**Figure 1 ijms-19-04119-f001:**
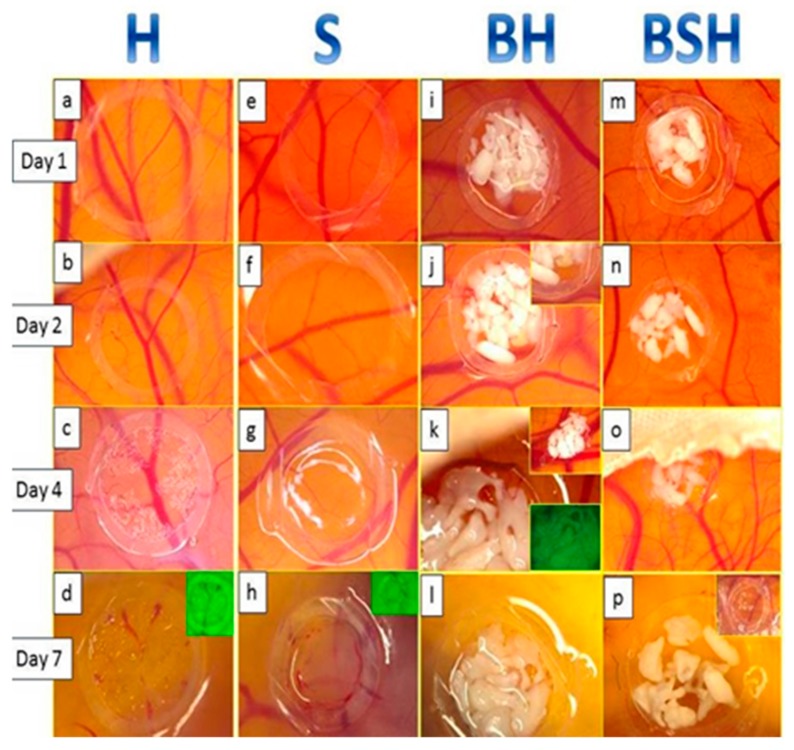
Stereomicroscopic view of CAM treated with hyaluronic acid (H), saline solution (S), bone substitute resuspended in H (BH), or in a mixture of S and H (BSH). A slightly vascular reaction was induced by H on day 1 and 2 (**a**,**b**), which was persistent until the end of the experiment (**c**,**d**). In contrast, S did not induce an increase in CAM blood vessel density (**e**–**h**). The BH complex applied on day 1 (**i**) dynamically changed its structure during the experiment. Small bone particles dispersed on H on day 2 (**j**) became structured in bigger bone lamellae on day 4 (**k**), and formed a compact mass by the end of the experiment (**l**). Compared with the BH complex, bone substitutes resuspended in a 1:1 mixture of H and S showed a less compact structure on CAM, with a restorative process starting from day 2 (**n**), and showing discohesive particles on day 4 and 7 (**o**,**p**).

**Figure 2 ijms-19-04119-f002:**
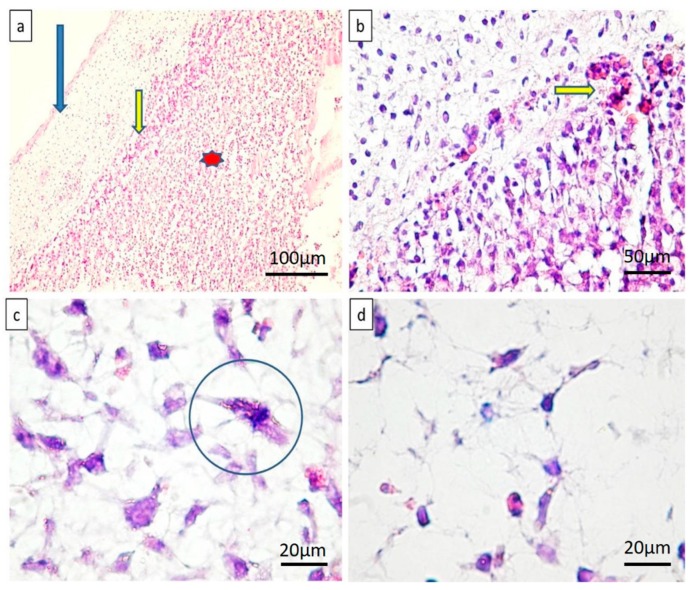
Microscopic view of CAM with a BH implant. There is an obvious integration of the BH implant inside CAM mesenchyme, with a clear stratification of the implant. The BH complex had a stratified appearance (**a**) below the CAM (blue arrow), the outer layer of BH (yellow arrow), and the internal area of BH (marked by red star). A well defined vascular network of small blood vessels penetrating the implant was observed (**b**) (yellow arrow). Stromal densification (**c**) compared to normal CAM (**d**). Mitotic activity of stromal mesenchymal cells was often seen amongst cells of CAM chorion, as marked in (**c**) (blue circle).

**Figure 3 ijms-19-04119-f003:**
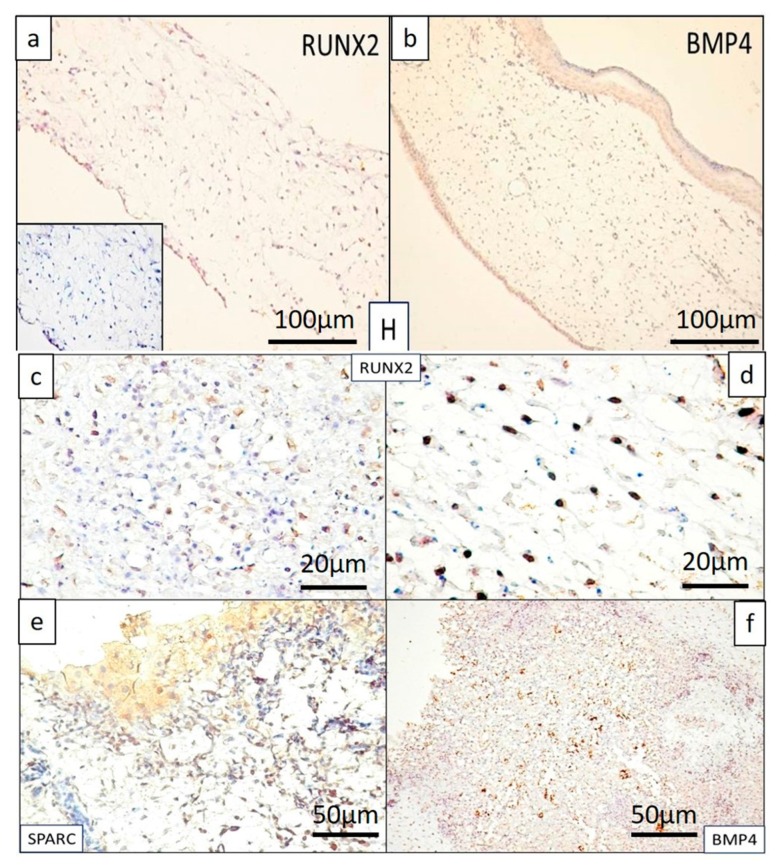
RUNX2 (**a**) and BMP4 (**b**) in hyaluronic acid (H) treated CAM specimens were negative. RUNX2 with moderate and heterogeneous expression in the BSH implant (**c**) and the implant adjacent to the CAM stroma (**d**). Moderate reaction for SPARC with an intense positive reaction in endothelial cells associated with inflammation adjacent to the implant (**e**). BMP4 is strongly positive in the vessels that invaded the BSH implant, but absent in the resuspended cells (**f**).

**Figure 4 ijms-19-04119-f004:**
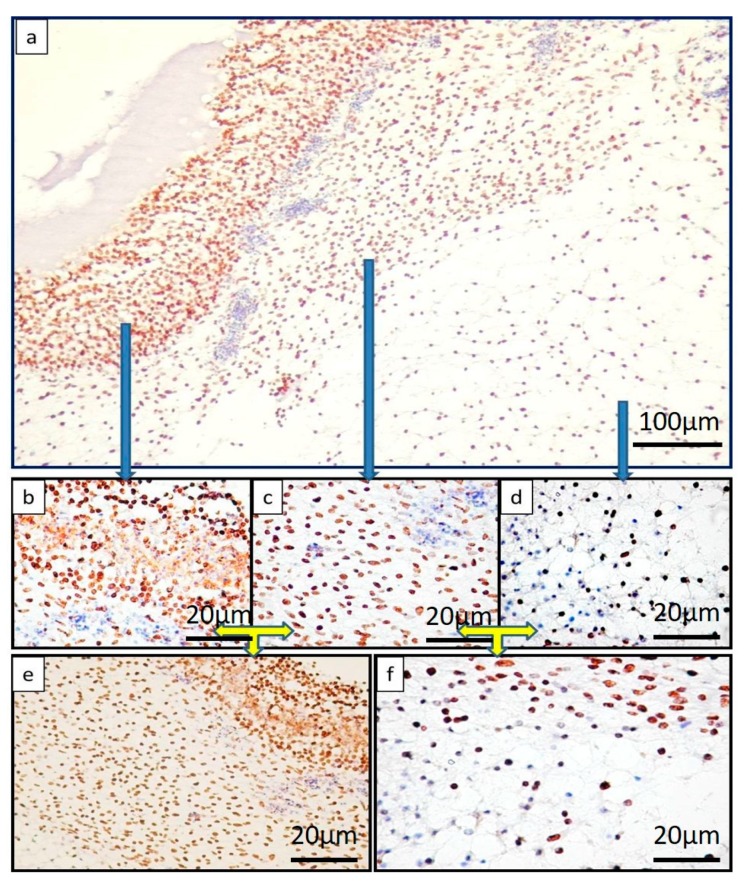
RUNX2 expression in BH treated samples, showing clearly the stratification of RUNX2 positive cells on distinct layers inside the CAM (**a**). RUNX2 positive cells (with nuclear pattern) had the highest density in the superficial layer (**b**) (with an average of 100 positive nuclei/field ×400 magnification), while layers below showed medium (**c**) (average of 70 positive nuclei/field ×400 magnification) and low density (**d**) (average of 50 positive nuclei/field ×400 magnification). Sharp limits in between layers can be shown based on the density of positive nuclei (**e**) between the superficial and middle layer, and (**f**) between the middle and the deep layer. Blue arrows indicate the detailed images from each three layers of BH treated samples. Yellow arrows show the limits between two consecutive layers and also highlighting differences regarding density of positive signals in between layers.

**Figure 5 ijms-19-04119-f005:**
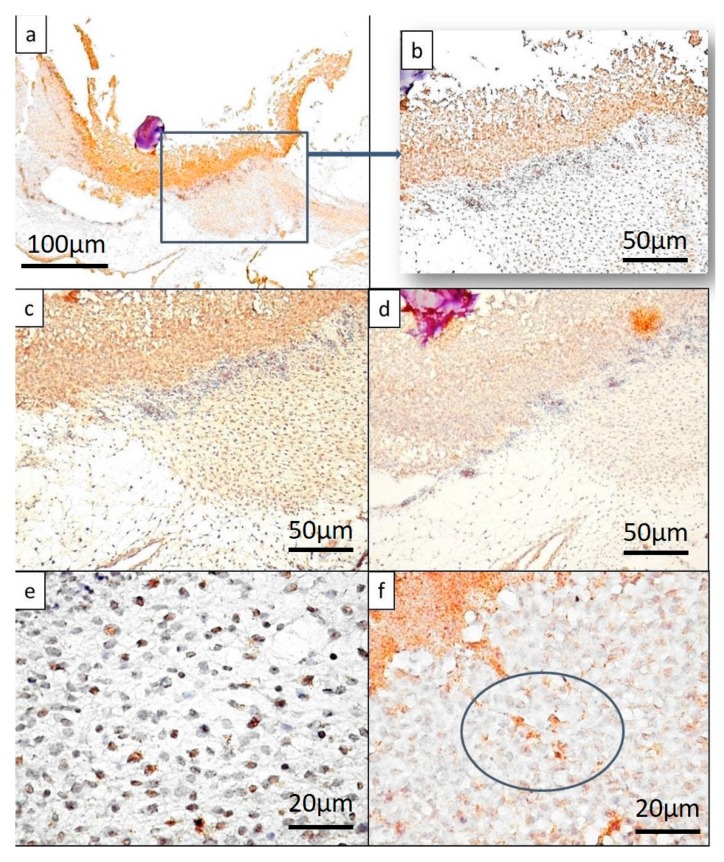
SPARC (**a**) and BMP4 (**d**) expression inside cellular components of CAM in BH treated specimens. Blue square indicates a suggestive zone for SPARC immunostaining, selected to be shown in detail in figure (b). Note that, stratification was also present for SPARC (**b**,**c**), as we have previously shown for RUNX2 in BH treated specimens. BMP4 was less pronounced in the densified chorion around the implant (**d**), and heterogeneously distributed in the nucleus (**e**) and cytoplasm ((**f**)**,** cells inside blue circle) of CAM mesenchymal cells for BH treated specimens where inflammation was absent.

**Table 1 ijms-19-04119-t001:** A comparative overview of the effects of bone substitutes and different resuspension media on CAM.

Substance/Sample	Angiogenic Response	Inflammation	RUNX2	SPARC	BMP4	Bone Particle Architecture
**H**	Present	No	Negative	Negative	Negative	-
**S**	No	Present	Negative	Negative	Negative	-
**BH**	Present, intense around implant	No	Positive with specific multilayered distribution. High density and intensity	Positive, respecting the layered distribution	Positive, heterogeneous distribution	Compact layer of interlaced bone lamellae, no resorption detected at the end of the experiment
**BS**	No	Present	+/−, inconsistent	Negative	Negative	Totally resorbed at the end of the experiment
**BSH**	Present, related to inflammation	Large inflammatory infiltrate	Positive	Positive in inflammatory cells, not in mesenchymal cells of CAM	Positive in inflammatory cells, not in mesenchymal cells of CAM	Dispersed arrangement of bone particles

**Table 2 ijms-19-04119-t002:** Composition and proportion of materials and substances applied on chick embryo chorioallantoic membrane (CAM).

Substance/Sample	BIO-GEN	Saline	Hyaluronic Barrier Gel	Saline + Hyaluronic
**H**	-	-	0.1 μL	-
**S**	-	0.1 μL	-	-
**BH**	0.5 g	-	0.5 mL	-
**BSH**	0.5 g	1 mL	0.3 mL	1 + 0.3 mL

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
