# Peer review of "Hyaluronic Acid/Bone Substitute Complex Implanted on Chick Embryo Chorioallantoic Membrane Induces Osteoblastic Differentiation and Angiogenesis, but not Inflammation"

_ijms, 2018, doi:10.3390/ijms19124119_

Reviewer 1 Report

1. Authors should place high resolution images. 

2. In Figure 4a, what is the orange circle?

3. Authors might be need to perform quantitative analysis of inflammation based on Figure 6 and 7.

Author Response

 Authors should place high resolution images. 

The images were replaced with high resolution ones. Thank you for your suggestion

2. In  Figure 4a,  what is the orange circle?

In previous Figure 4a, now 2a, there was no orange circle. it was a red star highlighting the inner layer of stratified implant. NOw, the red star was explained in the legend of the figure.

3. Authors might be need to perform quantitative analysis of inflammation based on Figure 6 and 7.

There is no inflammation in figures previously numbered as 6 and 7, now been numbered as 4 and 5 in the revised version. RUNX2, SPARC and BMP4 were assessed in CAM cells, highlighting differentiated cells through osteoblastic lineage. There is no inflammation because it is about BH implant which did not induced inflammation as has been stated in the body of the Results section. Therefore, for a better understanting the legend of these 2 figures was extensively revised. Thus, quantitative analysis of inflammation is not relevant for the paper.

Immunohistochemistry is a semi quantitative method so, in this moment we inserted inside the legend of figure 4 an average number of positive nuclei to highlight differences between layers regarading the density of RUNX2 positive cells important for the present study

English was extensively revised by a native speaker. Thank you.

Attached you have the revised version of the manuscript. Kind regrds and thank you for suggestions. Sincerely, AMCimpean

Reviewer 2 Report

- every explanation underneath the image must be clearer and fully explain the image.

- Figure 2 till Figure 7 must have a histological measure

- Authors should place high resolution images.
- And also be careful with form of the picture, to not stretch to much...like in picture 3
- In Figure 4a, what is the orange box
- All pictures has to have histology measure
- Authors might be need to perform quantitative analysis of inflammation with histomorphometry
- After qualitative histology explanation we need quantitative result

Author Response

Dear reviewer,

Thank you so much for the suggestions.

We consider that there were to many images and we remaind with 5 out of 7 images in the paper. All of them have in this momet histological measure as you mention (scale bar). 

High resolution images were added instead of previous ones.

In the pictures 6 and 7 we have no inflammation. Thus, an extensive, more clear explanations were added to each picture. The aim of the paper is not to assess inflammation and thus we consider that quantitative assessment of it is not our purpose and is not relevant for the paper. Moreover, on chick embry chorioallantoic membrane model known as a model where inflammation is absent in normal conditions (because CAM has no immune system) just to mention that we have or ot inflammation is enough for the purpose of our study. When we have inflammation after alveolar ridge augmentation, this augmentation is failed and in this context the quantification of inflammation by quantitative methods is not relevant. 

1.     There is no inflammation in figures previously numbered as 6 and 7, now been numbered as 4 and 5 in the revised version. RUNX2, SPARC and BMP4 were assessed in CAM cells, highlighting differentiated cells through osteoblastic lineage. There is no inflammation because it is about BH implant which did not induced inflammation as has been stated in the body of the Results section. Therefore, for a better understanting the legend of these 2 figures was extensively revised. Thus, quantitative analysis of inflammation is not relevant for the paper.

2.     Immunohistochemistry is a semi quantitative method so, in this moment we inserted inside the legend of figure 4 an average number of positive nuclei to highlight differences between layers regarading the density of RUNX2 positive cells important for the present study

3.     English was extensively revised by a native speaker. Thank you.

You have attached the revised version of the manuscript. Thanks a lot. AMCimpean

Round  2

Reviewer 1 Report

Almost part of this manuscript is acceptable.

However, in Fig. 2a, I noticed a strange orange artifical point.

Author should describe the detail of it or replace this figure.

Author Response

Dear reviewer,

Thank you for your previous suggestions. Regarding the orange circle in figure 2a, I would like to mention that no circle is added in figure 2a. The only orange element in the figure 2a is a...red star applied on the deepest 3rd layer of BH complex induced stratification instead of a third arrow which could somehow impare the microscopic image of stratification of the deepest layer (if I would add a third arrow the microscopic image would not be relevant). Thus I added a red star (the only red-orange element from the picture) to highlight this third layer and this red star is explained in the legend of the figure in the revised version of the manuscript. If on your draft appear other orange circle or element please mark it in order to be seen by me. In the version I attach as Revision 2 I highlighted in green , in the legend of figure 2 where I inserted explanation for red star.

With all my gratitude,

AMCimpean